# Machine learning-based models to predict the conversion of normal blood pressure to hypertension within 5-year follow-up

**Aref Andishgar**[1], **Sina Bazmi**[2], **Reza Tabrizi**[3]*, **Maziyar Rismani**[2],
**Omid Keshavarzian**[4], **Babak Pezeshki**[5], **Fariba Ahmadizar**[6]

**1** USERN Office, Fasa University of Medical Sciences, Fasa, Iran, **2** Student Research Committee, Fasa University of Medical Sciences, Fasa, Iran, **3** Noncommunicable Diseases Research Center, Fasa University of Medical Science, Fasa, Iran, **4** School of Medicine, Shiraz University of Medical Sciences, Shiraz, Iran, **5** Clinical Research Development Unit, Valiasr Hospital, Fasa University of Medical Sciences, Fasa, Iran, **6** Department of Data Science and Biostatistics, Julius Global Health, University Medical Center Utrecht, Utrecht, The Netherlands

* kmsrc89@gmail.com

**Data Availability Statement:** In our institutional policy, it is not stated that the data should be made public, and a data and material transfer agreement

## Abstract

### Background

Factors contributing to the development of hypertension exhibit significant variations across countries and regions. Our objective was to predict individuals at risk of developing hypertension within a 5-year period in a rural Middle Eastern area.

### Methods

This longitudinal study utilized data from the Fasa Adults Cohort Study (FACS). The study initially included 10,118 participants aged 35–70 years in rural districts of Fasa, Iran, with a follow-up of 3,000 participants after 5 years using random sampling. A total of 160 variables were included in the machine learning (ML) models, and feature scaling and one-hot encoding were employed for data processing. Ten supervised ML algorithms were utilized, namely logistic regression (LR), support vector machine (SVM), random forest (RF), Gaussian naive Bayes (GNB), linear discriminant analysis (LDA), k-nearest neighbors (KNN), gradient boosting machine (GBM), extreme gradient boosting (XGB), cat boost (CAT), and light gradient boosting machine (LGBM). Hyperparameter tuning was performed using various combinations of hyperparameters to identify the optimal model. Synthetic Minority Over-sampling Technology (SMOTE) was used to balance the training data, and feature selection was conducted using SHapley Additive exPlanations (SHAP).

### Results

Out of 2,288 participants who met the criteria, 251 individuals (10.9%) were diagnosed with new hypertension. The LGBM model (determined to be the optimal model) with the top 30 features achieved an AUC of 0.67, an f1-score of 0.23, and an AUC-PR of 0.26. The top three predictors of hypertension were baseline systolic blood pressure (SBP), gender, and

should not allow further transfer of data without the provider's prior written consent. However, the data can be made available upon request from the corresponding author, who is a member of this team. Additionally, the dataset generated for this study is available upon request to the Fasa Non-Communicable Diseases Research Center management team. They can be contacted via telephone at +987153314068 or via email at ncdrc.fums.ac.ir@gmail.com.

**Funding:** The author(s) received no specific funding for this work.

**Competing interests:** The authors have declared that no competing interests exist.

waist-to-hip ratio (WHR), with AUCs of 0.66, 0.58, and 0.63, respectively. Hematuria in urine tests and family history of hypertension ranked fourth and fifth.

## Conclusion

ML models have the potential to be valuable decision-making tools in evaluating the need for early lifestyle modification or medical intervention in individuals at risk of developing hypertension.

## Introduction

Hypertension, a prevalent chronic multifactorial disease, remains a significant challenge in the modern world [1]. In 2021, the World Health Organization estimated that approximately one-third of the global population have hypertension, two-thirds of those found in low- and middle-income countries [2]. Despite advancements in diagnosis and treatment, the prevalence of hypertension in these countries continues to rise [3]. Iran, for instance, reports a 25% prevalence of hypertension [4]. According to World Health Organization reports, hypertension contributes to an annual toll of 9.4 million deaths. In low- and middle-income countries, hypertension was responsible for approximately 8.5 million deaths in 2015, accounting for 88% of global hypertension-related mortality [5]. Specifically, hypertension stands as a primary cause of mortality in the Middle East [6]. Referred to as a silent killer, hypertension becomes apparent only at hazardous pointes, leading to events such as heart attacks or strokes [7]. Despite being controllable through cost-effective medications and timely interventions [8], many hypertensive patients remain undiagnosed due to insufficient awareness of screening and risk factors [9]. Moreover, care episodes for hypertension in low- to middle-income countries incur costs ranging from $500 to $1500, with monthly treatment expenses averaging around $22 [10], and hypertension commonly develops among middle-aged individuals, impacting productivity and imposing additional burdens on economic systems [11]. Given the high level of costs and complications associated with the chronic disease, studies have aimed to estimate hypertension risks for more effective prevention and management of complications [1]. Among the most renowned risk assessment tools is the Framingham Risk Score for predicting cardiovascular diseases [12]. However, these models lack sufficient diversity in encompassing different ethnicities, necessitating the constant development of tailored risk prediction models for specific populations [13].

Machine learning (ML), an integral component of artificial intelligence (AI), has gained significant traction in recent years due to its superior performance in risk classification tools compared to conventional statistical techniques [14]. This technology enables computers to learn without direct programming and adeptly analyze intricate data interactions [15]. Typically, ML surpasses traditional statistical methods by reducing bias, autonomously handling missing variables with minimal intervention in original data, managing distorted variables, and ensuring balanced data, thereby yielding superior outcomes [15]. Furthermore, ML models have the capacity to represent nonlinear relationships and enhance overall predictive accuracy [16]. Consequently, ML methods serve as a valuable tool for automating disease prediction [15].

While the precise origins of hypertension remain elusive, factors such as genetics, excessive salt intake, reduced physical activity, and being obese are known contributors to its progression [17]. These and variables such as educational levels and income, among others, exhibit

significant variations across countries and regions [8], underscoring the need for further research to develop location-specific risk assessment tools. Numerous studies have sought to predict hypertension using AI-based ML models. However, the data from these studies have been primarily cross-sectional, and there is no evidence indicating the successful implementation of these algorithms in clinical settings in the rural Middle East areas. Additionally, to date, no longitudinal hypertension prediction model has been established on the total population of these regions.

In this investigation, we aim to assess and contrast the efficacy of various ML methods utilizing a longitudinal rural middle eastern dataset to forecast individuals susceptible to developing hypertension within a 5-year span, hence identifying individuals with a higher probability of benefiting from treatments. We scrutinize and compare ten ML techniques to derive the optimal model for predicting hypertension risk. The assessment of models involves multiple metrics, employing a range of validation techniques and evaluation criteria.

## Methods

### 1. Data source

This is a retrospective longitudinal study based on the Fasa Adults Cohort Study (FACS) data. FACS study has 10 118 participants aged 35–70 years in Sheshdeh and Qarabolagh districts of Fasa city. FACS was created to assess the risk factors that predispose Fasa's rural residents to Non-Communicable Diseases (NCDs), including cardiovascular diseases. FACS enrollment began in October 2014 and ended in September 2016 in an area with 84% rural residents. Since September 2021, when the fourth follow-up was completed, the cohort study has entered the re-evaluation phase of the same variables as the registration phase, with 3000 of the first phase participants scheduled to participate. Random sampling was used to select participants for this phase of the study. The re-evaluation phase includes all of the steps taken, clinical examinations performed, biological samples taken, and questionnaires administered during the registration phase [18].

### 2. Study population

Our study sample is selected as a census from the FACS. Five inclusion criteria were considered to include people in the study: 1. Participants with 5 years of follow-up 2. Participants with 5 years data available 3. Participants without hypertension diseases at the first phase (with the same diagnostic criteria mentioned in the final outcome section) 4. Being alive at the end of the follow-up. Finally, 2288 participants were included with census method. The study steps are summarized in a flowchart displayed in Fig 1.

### 3. Data preparation and preprocessing

Most of the variables had no missing data and the other had < 10% missing data. For continuous variables, mean imputation was employed, while for categorical variables, median imputation was used to replace missing data. Finally, a total of 160 variables were included in the ML models. The list of all variables is included in S1 Table. Moreover, data must be processed before using ML models. Two methods were employed in this process: feature scaling and one-hot encoding. They were used to process the continuous variables and variables with more than 2 categories, respectively. In this study, standard scaling procedure was used which transforms continuous variables with a range from -1 to +1. One-hot encoding was applied to produce dummy variables which takes only the value of 0 or 1.

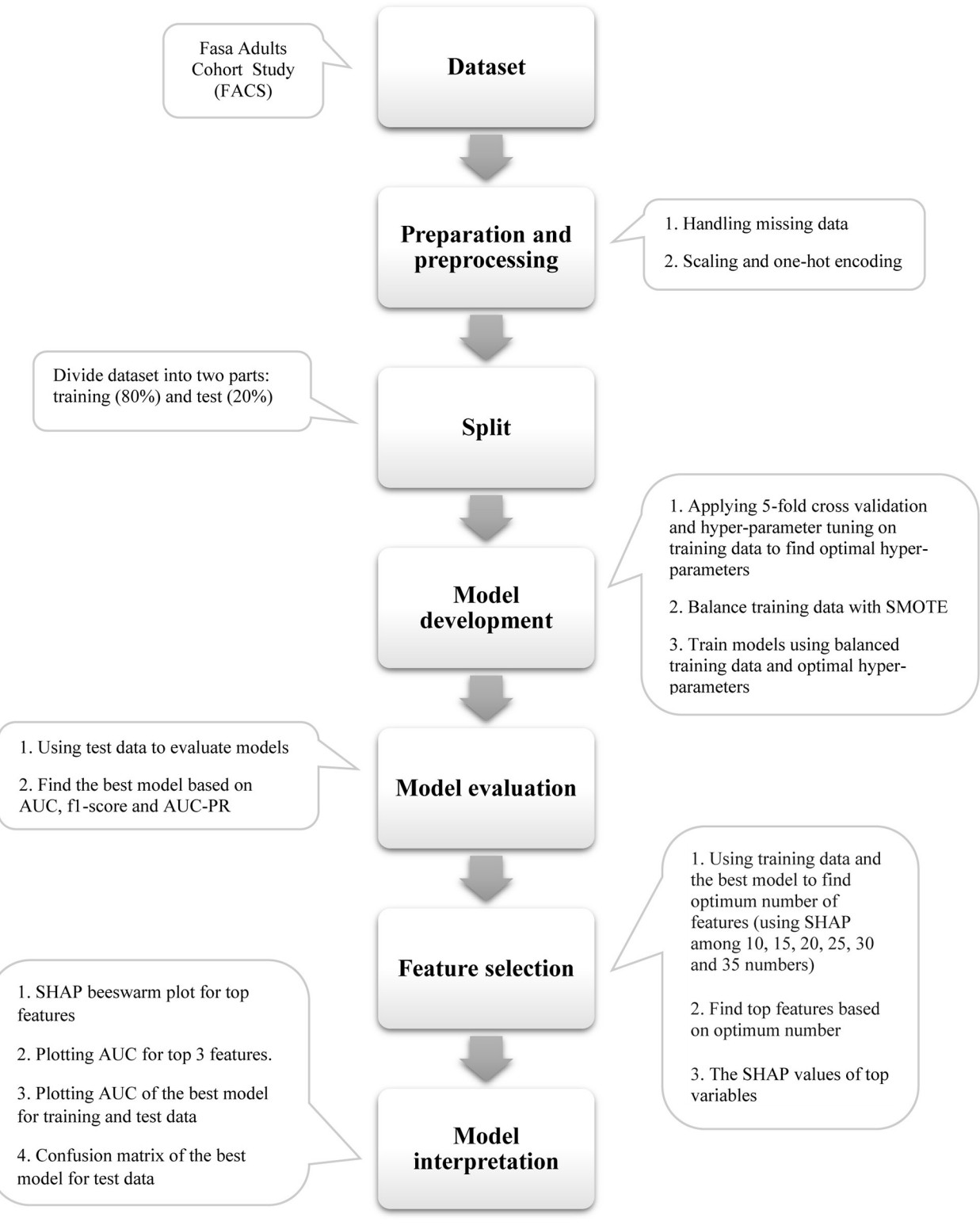

**Fig 1. Flowchart of this study.**

## 4. Final outcome

In the FACS study, a person was diagnosed with hypertension if they had systolic blood pressure (SBP) $\geq$140 mmHg or diastolic blood pressure (DBP) $\geq$90 mmHg on at least two episodes (15 minutes apart), or consuming anti-hypertensive drugs due to previous diagnosis [17]. In the current study, having hypertension after 5 years of follow-up analyzed as a classified outcome (hypertensive participants / non-hypertensive participants).

## 5. Splitting data

To avoid overfitting, the dataset was divided into two parts: training (80%) and test (20%) data. Training was used for training the models, hyper-parameter tuning and 5-fold cross validation. Test data was blind to the training data and was used for final evaluation and internal validation of the ML models. Training and test data was followed 5 years until final outcome was achieved (Fig 2).

## 6. Machine learning algorithms

In this study, ten supervised ML algorithms were used: logistic regression (LR), support vector machine (SVM), random forest (RF), gaussian naive bayes (GNB), linear discriminant analysis (LDA), k-nearest neighbors (KNN), gradient boosting machine (GBM), extreme gradient boosting (XGB), cat boost (CAT) and light gradient boosting machine (LGBM). We used a variety of ML methods to make sure the dataset was thoroughly explored. Every algorithm

**Fig 2. Procedure of splitting dataset into training (80%) and test (20%) parts.**

possesses distinct advantages and disadvantages, and our objective was to evaluate each one's performance independently in several research domains. LR was implemented for its simplicity. SVM was selected for its capability in handling high-dimensional data and finding complex relationships. RF was leveraged as an ensemble learning model. GNB provided a computationally efficient approach. LDA offered interpretability. KNN was implemented to detect local patterns. GBM sequentially refined model performance. XGB can perform with high accuracy in large datasets. CAT optimized categorical feature handling, and LGBM efficiently managed larger datasets with swift training. This multifaceted strategy sought to capitalize on the distinct advantages of every model, guaranteeing a thorough examination of the dataset. By not depending only on a single model, we were able to prevent any bias and obtain a comprehensive comprehension of the data.

Anaconda (version 4.12.0) on the Visual Studio Code Platform (version 1.76.2) and python (version 3.9.12) was used to implement all ML algorithms. Furthermore, the machine algorithms were run using the Scikit-Learn Module (version 1.1.3) [19].

## 7. Model development

At first, 5-fold cross validation and hyper-parameter tuning were applied on training data to find the optimal hyper-parameters. In this stage all features were used. The 5-fold approach separated all of the training data into 5 equal parts, and each time one of the parts was considered validation data, it trained itself and reported the accuracy, and eventually, the average of all 5 accuracies was obtained. Each ML model's accuracy may now be changed by adjusting its hyper-parameters. Various combinations of hyper-parameters were utilized in the hyper-parameter tuning process to find the best combination of hyper-parameters. For the hyper-parameter tuning step, the grid search approach was employed [20] (S2 Table).

Second, over-sampling was employed to balance the outcome values. The Synthetic Minority Over-sampling Technology (SMOTE) was used to balance the training data. This technique oversamples the minority group by creating "fake" instances. SMOTE selects samples from the minority class and creates "fake" samples along the same line segment, linking some or all of the k nearest neighbors of the minority class [21]. Hypertensive participants were the minority class and SMOTE generated 1428 instances to equalize hypertensive and non-hypertensive individuals.

Finally, ML models were trained using balanced training data and the best hyper-parameters.

## 8. Model evaluation

All trained ML models were applied to test data. For the final evaluation and comparison of the ML models, three metrics were used: Area under receiver operating characteristic curve (AUC), f1-score and area under the precision-recall curve (AUC-PR) (Fig 3A, S3 Table). In addition, for more detail, S3 Table includes measures such as accuracy, sensitivity, and specificity. The following equations are used to determine the evaluation metrics:

**Accuracy** $= (TP + TN)/(TP + FP + TN + FN)$

**Sensitivity** $= TP/(TP + FN)$

**Specificity** $= TN/(TN + FP)$

**F1-score** $= 2*TP/(2*TP + FP + FN)$

TP stands for true positive rate, TN is for true negative rate, FP stands for false positive rate, and FN stands for false negative rate. Finally, LGBM model was chosen as the best ML model based on AUC, f1-score and AUC-PR.

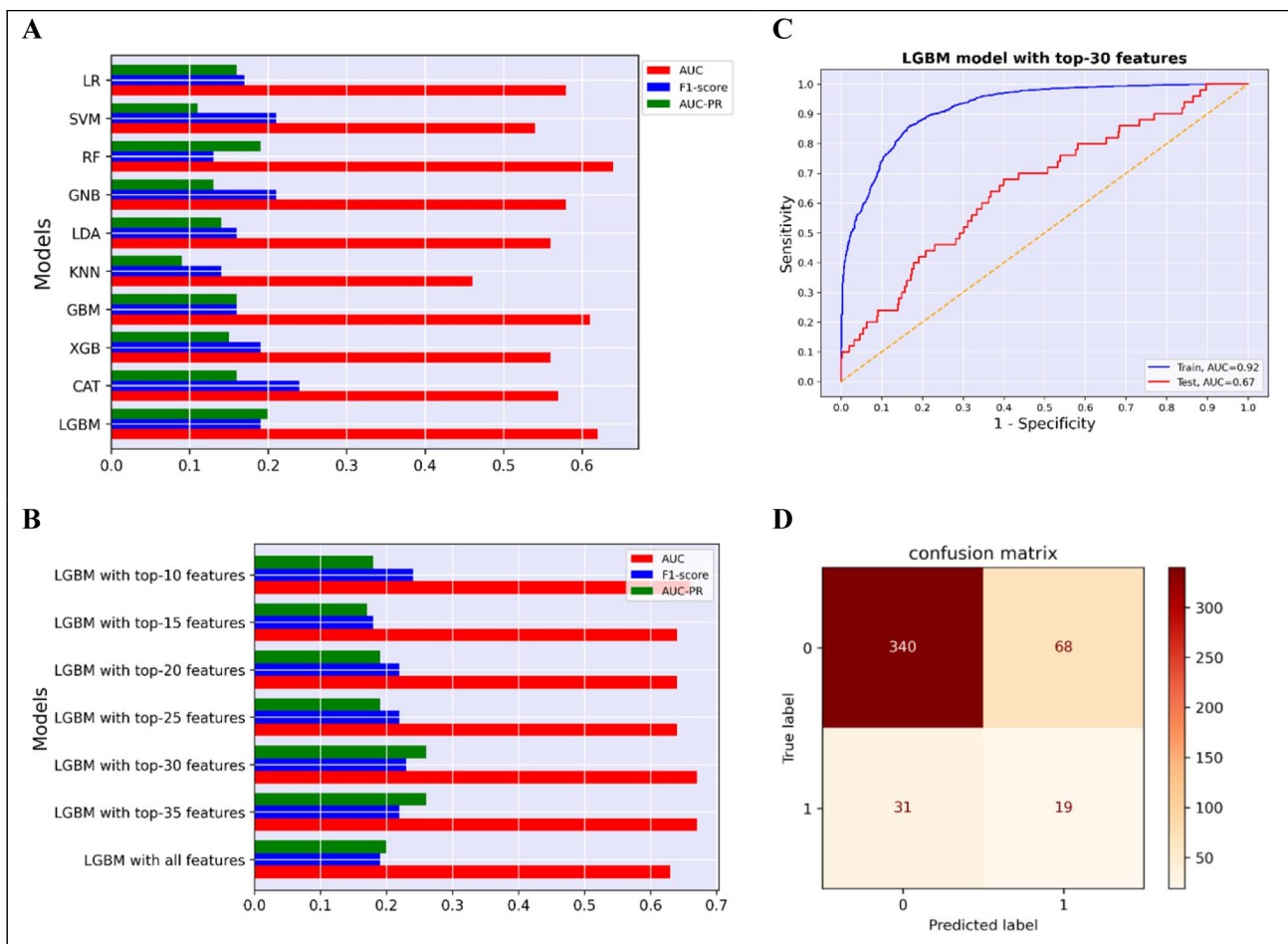

**Fig 3. Comparative analysis of model performance indicators and diagnostic visualizations for full-featured and simplified models across ten algorithms.** (A), Comparison the AUC, f1-score and AUC-PR among the models with ten algorithms using all features. (B), Comparison of the AUC, f1-score and AUC-PR among simplified models and the model with all features. (C-D), ROC curve and confusion matrix of the LGBM model with top-30 features. LR; Logistic Regression, SVM; Support Vector Machine, RF; Random Forest, GNB; Gaussian Naive Bayes, LDA; Linear Discriminant Analysis, KNN; K-Nearest Neighbors, GBM; Gradient Boosting Machine, XGB; Extreme Gradient Boosting, CAT; Cat boost, LGBM; Light Gradient Boosting Machine, AUC; Area Under the ROC Curve, ROC; Receiver operating characteristic, AUC-PR; Area Under the Precision-Recall curve.

## 9. Feature selection

To accomplish efficient data reduction, feature selection approaches can be utilized. This is helpful in identifying more accurate ML models and reduce computational costs. There are three types of feature selection: wrapper, filter, and embedded methods [22]. SHapley Additive exPlanations (SHAP) was used as a wrapper method. SHAP is a uniform way to explaining any ML model's output. It combines game theory to local explanations, bringing together various earlier approaches and represents the only consistent and locally correct additive feature attribution method based on expectations. It has become a feature selection method in the recent years [23, 24].

Then, SHAP and LGBM models were combined to determine the optimal amount of features between 10, 15, 20, 25, 30, and 35. The best performance was achieved by a subset of 30 characteristics (Fig 3B, S4 Table). Fig 4B shows the top 30 features and their importance in predicting hypertension.

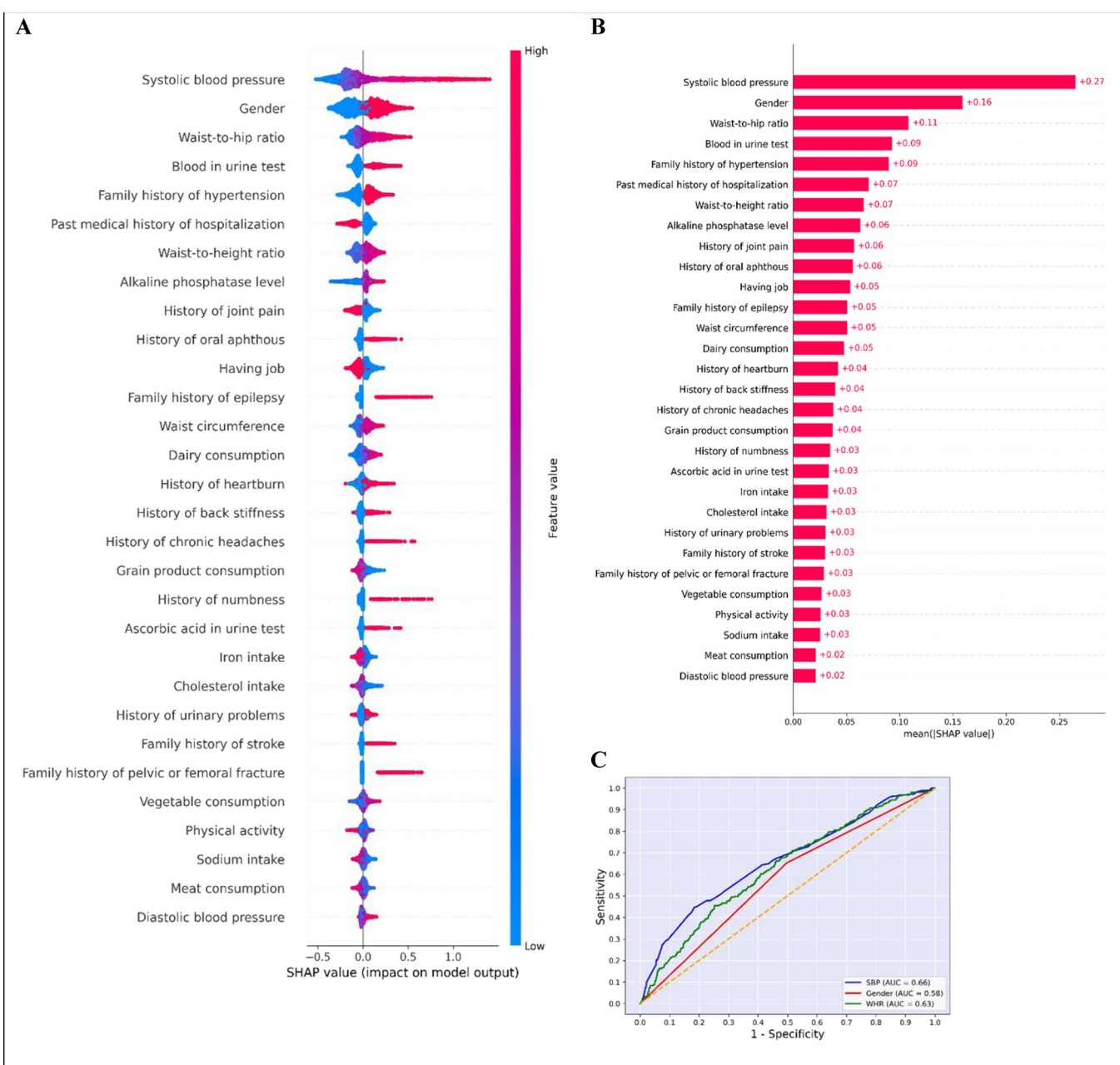

**Fig 4. Interpret LGBM model with top-30 features and its performance.** (A), SHAP beeswarm plot for top features. The plot below sorts features by the sum of SHAP value magnitudes over all samples, and uses SHAP values to show the distribution of the impacts each feature has on the model output. The color represents the feature value (red high, blue low). This reveals for example that a high systolic blood pressure highers the predicted home price. (B), The SHAP values of top-30 variables. The input features on the y-axis are arranged in descending importance, and the values on the x-axis represent the mean influence of each feature on the size of the model output based on SHAP analysis. (C), Receiver operating characteristic (ROC) curves of top-3 features. LGBM; Light Gradient Boosting Machine, AUC; Area Under the ROC Curve, ROC; Receiver operating characteristic, SHAP; Shapley Additive exPlanations, SBP: Systolic Blood Pressure, WHR; Waist-to-Hip Ratio.

Table 1 displays a descriptive analysis of chosen characteristics. Statistical tests such as Independent Samples Test, Chi-Square Test, and Mann-Whitney Test were utilized. Statistical significance was defined as P-values less than 0.05. The data was analyzed using SPSS version 18 (IBM Corp., Armonk, N.Y., USA).

**Table 1. General and top-30 important characteristics of the participants according to hypertension after 5 years of follow-up (Total number of participants = 2288).**

| Variables | | Hypertension | | P-value |
|---|---|---|---|---|
| | | No (N = 2037) | Yes (N = 251) | |
| Age, years* | | 48.60±8.21 | 49.88±8.13 | 0.020[a] |
| Sex | Male | 1023 (50.2) | 87 (34.6) | ≤0.001[b] |
| | Female | 1014 (49.8) | 164 (65.4) | |
| Physical activity, MET | | 42.99±11.93 | 40.66±8.82 | ≤0.001[a] |
| Systolic blood pressure, mmHg | | 105[93,117] | 113[100,126] | ≤0.001[c] |
| Diastolic blood pressure, mmHg | | 71[62,80] | 74[65,83] | ≤0.001[c] |
| Waist circumference, cm | | 91.97±11.06 | 96.87±11.61 | ≤0.001[a] |
| Waist-to-hip ratio | | 0.92±0.06 | 0.95±0.05 | ≤0.001[a] |
| Waist-to-height ratio | | 0.56±0.07 | 0.60±0.07 | ≤0.001[a] |
| Alkaline phosphatase level (ALP), U/L | | 204.40±79.69 | 210.45±65.47 | 0.248[a] |
| Ascorbic acid in urine test, yes | | 288(87.8) | 40(12.2) | 0.443[b] |
| Blood in urine test, yes | | 620(86.4) | 98(13.6) | 0.006[b] |
| Iron intake, mg/day | | 24.03±10.98 | 22.27±9.88 | 0.016[a] |
| Sodium intake, mg/day | | 4782.3±2061.5 | 4447.1±1702.8 | 0.004[a] |
| Cholesterol intake, mg/day | | 278.14±79.69 | 256.20±169.25 | 0.043[a] |
| Grain products consumption, gr/day | | 704.23±331.86 | 641.76±284.17 | 0.004[a] |
| Vegetable consumption, gr/day | | 587.97±321.82 | 610.04±321.96 | 0.305[a] |
| Dairy products consumption, gr/day | | 212.48±179.60 | 231.17±188.67 | 0.122[a] |
| Meat consumption, gr/day | | 99.35±63.85 | 91.39±64.25 | 0.063[a] |
| History of joint pain, yes | | 828(89.0) | 102(11.0) | 0.997[b] |
| History of oral aphthous, yes | | 403(85.9) | 66(14.1) | 0.016[b] |
| History of heartburn, yes | | 721(87.7) | 101(12.3) | 0.131[b] |
| History of back stiffness, yes | | 497(86.9) | 75(13.1) | 0.0.58[b] |
| History of chronic headaches, yes | | 259(84.6) | 47(15.4) | 0.008[b] |
| History of numbness, yes | | 139(85.3) | 24(14.7) | 0.112[b] |
| History of urinary problems, yes | | 880(88.4) | 116(11.6) | 0.364[b] |
| Past medical history of hospitalization, yes | | 662(93.0) | 50(7.0) | ≤0.001[b] |
| Family history of hypertension, yes | | 986(87.2) | 145(12.8) | ≤0.001[b] |
| Family history of epilepsy, yes | | 120(83.9) | 23(16.1) | 0.043[b] |
| Family history of pelvic or femoral fracture, yes | | 122(83.0) | 25(17.0) | 0.015[b] |
| Family history of stroke, yes | | 257(85.7) | 43(14.3) | 0.046[b] |
| Having job, yes | | 1107(91.8) | 99(8.2) | ≤0.001[b] |

Data was presented as Mean ± SD, Median [IQR], and Number (%). Statistical analyses such as a: Independent Samples Test, b: Chi-Square, and c: Mann-Whitney Test were used. MET; Metabolic Equivalent

*Age is not one of top-30 variables and it is only presented as one of the demographic variables of the study.

## 10. Model interpretation

ROC curve and confusion matrix of the LGBM model with top-30 features is displayed in Fig 3C & 3D. The SHAP analysis was utilized to comprehend the LGBM model. SHAP values for the top features were determined in detail (Fig 4B). The beeswarm plot is designed to display an information-dense summary of how the top features in a dataset impact the model's output (Fig 4A).

### 11. Ethics approval and consent to participate

Our study protocol was approved by the Fasa University of Medical Sciences Research Council and Ethics Committee (approval code: IR.FUMS.REC.1402.133) and adhered to Helsinki guidelines. Furthermore, all subjects provided written informed consent before participating. The authors have no access to information that could identify individual participants during or after data collection.

## Results

### 1. Compare performance of machine learning algorithms

Fig 3A and S3 Table display the performance of all ML models based on various metrics. To make the final decision and discover the best ML model, the AUC, f1-score and AUC-PR metrics were considered. The highest AUC was achieved by RF (0.65). GBM and LGBM were ranked second and third, respectively, with AUC = 0.63. Among the ML models, CAT obtained the highest f1-score. GNB and SVM came in second and third, with f1-scores of 0.21 and 0.20, respectively. LGBM had the highest AUC-PR (0.20), while RF was second (AUC-PR = 0.19). Ultimately, the optimal model was determined to be LGBM. At last, 30 of the greatest features were chosen as the best numbers for predicting hypertension with LBGM model. The LBGM with top-30 features had AUC = 0.67, f1-score = 0.23 and AUC-PR = 0.26.

### 2. Descriptive analytics of top-30 variables of participants

In this research, 251 people (10.9%) were diagnosed with hypertension (Table 1). Women were more likely than males to have hypertension (p-value<0.05). In hypertension individuals, SBP, DBP, waist-to-hip ratio, and waist-to-height ratio were higher (p-value<0.05). Physical activity, blood in urine test, iron intake, sodium intake, cholesterol intake, grain products consumption, meat consumption, history of oral aphthous, history of chronic headaches, past medical history of hospitalization, family history of hypertension, family history of epilepsy, family history of pelvic or femoral fracture, family history of stroke, and having a job were all higher in hypertensive patients (p-value<0.05). There were no statistically significant differences in alkaline phosphatase level, ascorbic acid in urine test, vegetable consumption, dairy products consumption, history of joint pain, history of heartburn, history of back stiffness and h7istory of urinary problems (p-value>0.05).

### 3. Feature importance

Fig 4B shows the top-30 features in order of importance. The top three predictors of hypertension were SBP, gender, and waist-to-hip ratio, with AUCs of 0.66, 0.58, and 0.63, respectively. Blood in urine test and family history of hypertension were in fourth and fifth rank, respectively. In Fig 4A features were ordered by their SHAP values and it shows a beeswarm plot. For better understanding of this plot, binary variables have values of zero and one, which one indicates a positive value. To interpret this plot, more SBP and waist-to-hip ratio increase the risk of having hypertension in the future. Female gender and the presence of blood in urine test increase the risk of having hypertension in the future.

## Discussion

This longitudinal study, based on the FACS with 10,118 participants aged 35–70, aimed to predict 5-year hypertension risk using ML. After selecting 2288 participants meeting specific criteria, 160 variables were processed and prepared for analysis. Various ML algorithms were employed, and Light Gradient Boosting Machine (LGBM) emerged as the optimal model. The

study eventually introduced the top 30 features, highlighting the top 5 factors of SBP, gender, waist to hip ratio, hematuria, and family history of hypertension significantly associated with hypertension development. The model achieved an AUC of 0.67.

So far, three relatively robust studies have been conducted to predict hypertension using ML in the Middle East. AlKaabi et al. [1] implemented three supervised ML algorithms in a cross-sectional study involving 987 individuals aged over 18 in Qatar, where the random forest model demonstrated the best performance with an AUC of 0.869. Our study had a much larger sample size, a stronger methodology, utilized more models and variables, and unlike this study, incorporated feature selection. Additionally, the study population wasn't entirely representative of the entire region, focusing solely on individuals residing in Qatar within a specific timeframe. Sakr et al. [6] conducted a longitudinal study on 23,095 suspected cardiovascular patients referred for exercise testing and followed up for 10 years, implementing six ML models. The RTF model achieved the best performance with an AUC of 0.93. This study was longitudinal, had a larger sample size, and attained a higher AUC. However, it did not encompass the general population, evaluating only patients referred for exercise testing, and focused mainly on factors related to cardiovascular diseases and exercise test results. Furthermore, Namatollahi et al. [25] designed a predictive model for hypertension based on factors associated with body composition in a cross-sectional study utilizing data from the same adult cohort in Fasa. This study also followed a cross-sectional design, focusing exclusively on factors related to body structure. In contrast, considering the follow-up phase of this cohort, we conducted the current longitudinal study, incorporating a broader range of factors. Most studies conducted in the field of ML models for predicting hypertension [8, 26–28], including studies by AlKaabi and Namatollahi [1, 25], were based on cross-sectional data. Firstly, cross-sectional studies cannot precisely determine the exact timing of future hypertension development in patients. Secondly, cross-sectional data often include numerous hypertension-related complications in patients' records, which essentially provide the ML models with an unfair advantage, artificially inflating their accuracy scores. This issue, known as data leakage, undermines the predictive reliability of the results, making them fundamentally non-generalizable to real-world scenarios. In contrast, longitudinal data, like ours, begins with patients who are initially healthy, showing no signs of hypertension or its extensive complications. Consequently, results derived from longitudinal data hold greater validity, and even lower scores are more valuable than the misleadingly elevated scores from cross-sectional models.

In our model, an interesting predictive factor that had less discussion in texts regarding its relation to hypertension was positive hematuria. Before this, only three studies [29–31] directly examined this connection, all exclusively on hemophiliac patients, a population with higher occurrences of hematuria and hypertension than the general population, and they were conducted with small sample sizes. Holme et al. [30] in their cross-sectional study did not find a significant correlation between the presence of hematuria and hypertension in these patients. Sun et al. [31], in their prospective study focusing solely on men, concluded that despite the high prevalence of hematuria and hypertension in hemophiliac patients, these two factors are not related, and hematuria is unlikely to lead to hypertension in the long term. Also, renal insufficiency in these patients in the follow-up was rare, questioning the renal damage as an intermediary for this relationship. However, this study was solely conducted on hemophiliac male patients and had a small sample size. Nonetheless, Qvistad et al. [29] in a recent study found that the connection between hematuria and hypertension becomes significant in patients with a family history of hypertension. Our study results were adjusted for a family history of hypertension, yet hematuria was selected as one of the top 5 predictive factors for hypertension in a 5-year model. Hematuria could be a sign of underlying kidney damage or dysfunction, which, although mild and overlooked, could, in the long term, alter blood

pressure regulation by affecting sodium balance, increasing fluid retention, and disrupting hormonal equilibrium, such as the renin-angiotensin-aldosterone system, ultimately leading to hypertension [32–34]. Additionally, factors causing hematuria might trigger an inflammatory response and endothelial dysfunction. If chronic, this inflammation and dysfunction could potentially increase vascular resistance, subsequently raising blood pressure and leading to hypertension [35]. Of course, both hypertension and hematuria share common risk factors such as obesity and smoking, but these factors are adjusted for in models. Longitudinal studies based on this hypothesis are needed to examine and confirm the relationship between hematuria and the likelihood of developing hypertension over time.

Repeatedly, anthropometric indices have been introduced as risk factors for cardiovascular diseases [36], and various studies have reported a strong correlation between WHR (waist-to-hip ratio) and hypertension. However, WHR as a specific predictor for the occurrence of hypertension has been less discussed. Initially, a cross-sectional study by Feldstein et al. [37] demonstrated that WHR might better and logically predict the risk of hypertension compared to other anthropometric indices. A meta-analysis of cross-sectional studies indicated that WHR is a better biomarker for cardiovascular diseases and hypertension risk [38]. Choi et al. in a large longitudinal study with a good sample size concluded that WHR has a significant and strong relationship with the occurrence of hypertension over time [39]. The use of WHR, compared to popular anthropometric indices like BMI and WC, could be more useful as it's easier to measure, doesn't have a linear relationship with other indices, and has shown consistency across different age and ethnic groups [40]. In our study, WHR was chosen as the third top predictive factor for hypertension in the next 5 years, aligning with the mentioned texts and similar studies.

Family history of hypertension, like other diseases, is associated with a higher chance of developing hypertension in an individual. Wang and colleagues' extensive 54-year longitudinal study on a cohort demonstrated that family history of hypertension, both from the father and mother, has an independent and strong correlation with the occurrence of hypertension over time [41]. Similarly, a recent longitudinal study by Kunnas et al. [42] with a 15-year follow-up and a more precise design showed similar results. In our study, a family history of hypertension was selected as the fifth top predictive factor for hypertension in the next 5 years, in line with the mentioned texts and similar studies.

The occurrence and prevalence of hypertension differ between men and women [43]. Generally, hypertension prevalence is usually higher in men than in women, but our model identified female gender as the second top predictive factor for hypertension in the next 5 years. Our study cohort included individuals aged 35 to 70 years. As age increases, especially beyond the sixth decade of life, the steepness of hypertension occurrence in women significantly rises [44]. Moreover, at older ages, specific hypertension risk factors for women, such as pregnancy-induced hypertension and menopause, become evident and prevalent, increasing the chances of developing hypertension at these ages [45]. Additionally, socioeconomically disadvantaged status is more associated with hypertension in women [46], which seems entirely logical given our study population in rural areas, predominantly with lower socioeconomic status. Considering that hypertension in women is a stronger risk factor for cardiovascular diseases [45], this result seems crucial.

Two factors, WHR and SBP, among the top predictive factors in our study, were in line with the top predictive factors in a similar and robust study conducted in Canada [47]. This conformity could indicate a percentage of similarity among different populations in predicting future hypertension.

Based on our ML model, individuals at high risk for developing hypertension can be recommended to modify their lifestyles and behaviors (such as physical activity, dietary changes,

smoking cessation, and alcohol consumption) to avoid hypertension and prevent all associated dangerous complications and costs [47]. It is further recommended to employ new ML models in various geographical regions where there is a wide diversity in hypertension risk factors, as each model may reveal new predictive factors for hypertension [28].

Our study had several strengths that set it apart from previous research. Firstly, it was a longitudinal study with a 5-year follow-up period, providing valuable insights into the long-term development of hypertension. Additionally, we employed feature selection approaches and utilized ten supervised ML algorithms, enhancing the robustness of our analysis. Furthermore, we conducted hyperparameter tuning to optimize the performance of our models. Moreover, our study has a significantly larger number and scope of variables compared to most studies conducted in this field, including the Canadian study [47].

Our study had a strong methodology; however, unfortunately, we faced severe data limitations. Out of the 3,000 followed individuals, only 251 developed hypertension. Due to this data limitation, our model's final F1 score was low, and the AUC did not reach a significantly high value. Despite the severe data limitations, we were able to achieve an AUC of approximately 0.67, which demonstrates the strength of our methodology. Upon the completion of the data collection for the Fasa cohort follow-up phase in the upcoming years, we will be able to enhance and fortify our models. Furthermore, we were unable to perform external validation with our models due to limitations in accessing complete datasets from different cohorts.

## Conclusion

ML models demonstrated effective performance in predicting hypertension and its related factors in our rural population. LGBM emerged as the optimal model. It eventually introduced the top 30 features, highlighting the top 5 factors of higher baseline SBP, female gender, higher WHR, positive hematuria, and family history of hypertension significantly associated with hypertension development in the future. The model achieved an AUC of 0.67, f1-score = 0.23 and AUC-PR = 0.26. Individuals identified as high risk can be recommended to modify their lifestyles and behaviors to prevent hypertension and associated complications and costs.

## Supporting information

**S1 Table. All characteristics and clinical features of participants.**
(DOCX)

**S2 Table. Finding the appropriate hyper-parameter values for each algorithm after hyperparameter tuning.** #Abbreviations, LR; Logistic Regression, SVM; Support Vector Machine, RF; Random Forest, GNB; Gaussian Naive Bayes, LDA; Linear Discriminant Analysis KNN; K-Nearest Neighbors, GBM; Gradient Boosting Machine, XGB; Extreme Gradient Boosting, CAT; Cat Boost, LGBM; Light Gradient Boosting Machine.
(DOCX)

**S3 Table. Performance of the ten machine learning algorithms using all features.** #Abbreviations, LR; Logistic Regression, SVM; Support Vector Machine, RF; Random Forest, GNB; Gaussian Naive Bayes, LDA; Linear Discriminant Analysis, KNN; K-Nearest Neighbors, GBM; Gradient Boosting Machine, XGB; Extreme Gradient Boosting, CAT; Cat boost, LGBM; Light Gradient Boosting Machine, AUC; Area Under the ROC Curve, ROC; Receiver operating characteristic, AUC-PR; Area Under the Precision-Recall curve.
(DOCX)

**S4 Table. Performance of the LGBM model with different number of features.** #Abbreviations, LGBM; Light Gradient Boosting Machine, AUC; Area Under the ROC Curve, ROC;

Receiver operating characteristic, AUC-PR; Area Under the Precision-Recall curve.
(DOCX)

## Acknowledgments

This project has been approved by the National Institutes for Medical Research Development (NIMAD), Tehran, Iran under code "4021292".

## Author Contributions

**Conceptualization:** Sina Bazmi, Reza Tabrizi, Maziyar Rismani.

**Data curation:** Aref Andishgar, Maziyar Rismani.

**Formal analysis:** Aref Andishgar.

**Investigation:** Sina Bazmi.

**Methodology:** Aref Andishgar, Maziyar Rismani.

**Project administration:** Reza Tabrizi.

**Resources:** Reza Tabrizi.

**Software:** Aref Andishgar.

**Supervision:** Reza Tabrizi, Omid Keshavarzian, Babak Pezeshki, Fariba Ahmadizar.

**Validation:** Aref Andishgar, Sina Bazmi, Omid Keshavarzian, Babak Pezeshki, Fariba Ahmadizar.

**Visualization:** Aref Andishgar.

**Writing – original draft:** Aref Andishgar, Sina Bazmi.

**Writing – review & editing:** Sina Bazmi.

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
