## [Decision Letter · Decision Letter 0]

29 Jan 2024

PONE-D-23-44016Machine learning-based models to predict the conversion of normal blood pressure to hypertension within 5-year follow-upPLOS ONE

Dear Dr. Tabrizi,

Thank you for submitting your manuscript to PLOS ONE. After careful consideration, we feel that it has merit but does not fully meet PLOS ONE’s publication criteria as it currently stands. Therefore, we invite you to submit a revised version of the manuscript that addresses the points raised during the review process.

The reviewers have raised several critical concerns regarding the manuscript. The authors are encouraged to address these issues which might need extensive changes. The English language of the manuscript should also be enhanced. 

We look forward to receiving your revised manuscript.

Kind regards,

Amir Hossein Behnoush

Academic Editor

PLOS ONE

Journal Requirements:

3. In this instance it seems there may be acceptable restrictions in place that prevent the public sharing of your minimal data. However, in line with our goal of ensuring long-term data availability to all interested researchers, PLOS’ Data Policy states that authors cannot be the sole named individuals responsible for ensuring data access (http://journals.plos.org/plosone/s/data-availability#loc-acceptable-data-sharing-methods).

Reviewers' comments:

Reviewer's Responses to Questions

**Comments to the Author**

1. Is the manuscript technically sound, and do the data support the conclusions?

Reviewer #1: Yes

Reviewer #2: Yes

Reviewer #3: Partly

2. Has the statistical analysis been performed appropriately and rigorously? 

Reviewer #1: Yes

Reviewer #2: Yes

Reviewer #3: Yes

3. Have the authors made all data underlying the findings in their manuscript fully available?

Reviewer #1: Yes

Reviewer #2: Yes

Reviewer #3: No

4. Is the manuscript presented in an intelligible fashion and written in standard English?

Reviewer #1: Yes

Reviewer #2: Yes

Reviewer #3: Yes

5. Review Comments to the Author

Reviewer #1: The study entitled "Machine learning-based models to predict the conversion of 1 normal blood pressure

2 to hypertension within 5-year follow-up" conducted by Andishgar et al. aimed to assess and contrast the efficacy of various machine learning methods evaluating individuals susceptibility to develop new onset hypertension within 5-years. the study is well-conducted, the aim is clear and authors developed the study in parallel of the main aim. however I have some major concerns regarding the method and results.

1-ML as a predictive tool should be in parallel with previous clinical findings. Top 30 important features seems not to be in agreement with some hypertension risk factors. in this study ALP level has much more predictive power than absence of "physical activity" which considers as a major risk factor of HTN.

2- authors reported the prevalence of hematuria in normal individuals 630 out of 2300. It is a huge number for the prevalence of this key feature.

3- I ask authors to add the logistic model, since this simple model showed a better performance than other models in many literatures.

Minor Comments:

1- in table 1, the percentage reported based on "row" as total, for example the proportion of male sex reported 92.2% in individuals W/O HTN, which is incorrect. this should be changed to "column" as total.

2- The conclusion should be according to the aim of the study, please add a sentence or two explaining the findings for best model.

Reviewer #2: The study titled "Machine learning-based models to predict the conversion of normal blood pressure to hypertension within 5-year follow-up" conducted by Andishgar and colleagues used ML models for prediction of hypertension. The study is well-designed. I have some major comments for improvement:

1- Abbreviations should be defined in their first use. Please ONLY use abbreviated forms after the definition (e.g., you defined ML several times).

2- The introduction is too long. Make it more concise.

3- Line 116: change "5" to "five"

4- Methods section 2: Have you excluded patients receiving anti-hypertensive drugs?

5- If possible, add external validation; else, mention it clearly in the discussion and limitations sections.

6- I found several typos and grammatical errors.

Reviewer #3: This manuscript comprehensively explores using machine learning (ML) techniques to predict hypertension risk in a rural Middle Eastern area. The study adopts a longitudinal design with an impressive initial sample size (10,118 participants) and follows up with 3,000 participants after five years. The insights into ML applications for hypertension risk prediction in a specific population are valuable, emphasizing the potential for early intervention. However, the moderate AUC suggests room for improvement, and the practical implementation of these models in healthcare would require further validation and consideration of real-world factors.

I have identified some points that could enhance the manuscript:

1. Lines 85-87: Clarify the term "few risk factors" by providing a specific range. Additionally, elaborate on "common machine learning approaches" by offering examples from previous studies.

2. Line 92: Introduce the acronym FACS before using it in this section for better reader understanding.

3. Lines 93-94: Distinguish between "common" and "established" machine learning techniques. Clarify how these terms differ and reconsider the use of "common" to avoid potential underestimation of previous approaches.

4. Line 108: Define the acronym NCDs before its use.

5. Line 108: Specify examples of "the most common ones" regarding NCDs for better context.

6. Line 117: Reevaluate the necessity of explicitly stating "Age between 35-70 years" as an inclusion criterion, given that it aligns with the larger study's age range.

7. Lines 123: Clarify the apparent conflict between the statement about missing data and the inclusion criterion.

8. Lines 123-124: Provide details on how the multiple imputation method was implemented. Specify if different ML models were trained on various imputed datasets and how this influenced later stages.

9. Lines 144-148: Explain the rationale behind choosing specific ML algorithms. Provide insights into why these algorithms were deemed suitable for the study.

10. Lines 156-158: Elaborate on your approach to combining hyperparameters. Explain whether a grid search or any specific method was used.

Addressing these points can improve the manuscript's clarity and give readers a more detailed understanding of the study's methodology and findings.

6. PLOS authors have the option to publish the peer review history of their article (what does this mean?). If published, this will include your full peer review and any attached files.

Reviewer #1: No

Reviewer #2: No

Reviewer #3: No

---

## [Author Response · Author response to Decision Letter 0]

3 Feb 2024

Reviewer #1

1. ML as a predictive tool should be in parallel with previous clinical findings. Top 30 important features seems not to be in agreement with some hypertension risk factors. in this study ALP level has much more predictive power than absence of "physical activity" which considers as a major risk factor of HTN.

Response:

We appreciate the reviewer's attention to detail. We also found this finding to be intriguing initially, and we believe it could be one of the significant contributions of our work to the existing literature. By identifying new potential risk factors, our study opens avenues for future research. Although our models did not exhibit the highest accuracies, we also observed that certain anthropometric features related to physical activity, such as waist-hip ratio and waist-to-height ratio, ranked higher than factors like ALP. Recent studies have emphasized the role of diet in the development of cerebrovascular diseases and diabetes, suggesting that it may be more influential than physical activity in these conditions. Furthermore, the association between ALP and hypertension has been discussed in the literature [1]. A prospective cohort study using data from the Kharameh cohort study, which is part of the Prospective Epidemiological Studies in Iran (PERSIAN) database, similar to ours, demonstrated that higher levels of ALP were associated with an increased risk of developing hypertension [2]. It is possible that this association is more pronounced genetically in Iran. Exploring its generalizability in future studies could be an interesting avenue to pursue. ALP has been suggested to be positively associated with hypertension due to its potential link to atherosclerosis and endothelial dysfunction [3]. Additionally, it has been proposed to be inversely related to endothelium-dependent vasodilation [4]. 

2. authors reported the prevalence of hematuria in normal individuals 630 out of 2300. It is a huge number for the prevalence of this key feature.

Response:

Thanks for bringing up this point. In our study, we used a validated dataset that was introduced and published in "International Journal of Epidemiology". We cited this dataset in our method section [5]. Additionally, the prevalence of hematuria we observed was 630 out of 2300 cases, which corresponds to 27 percent. This prevalence could be reasonable, considering that a review study has reported hematuria prevalence rates of up to 31 percent in various populations [6] (“The reported prevalence of asymptomatic microhematuria (aMH) ranges between 1.7% and 31.1%”)

3. I ask authors to add the logistic model, since this simple model showed a better performance than other models in many literatures.

Response:

As you said, we also added logistic regression model, as can be seen in the figures and the manuscript. It didn’t perform better than the LGBM model and our final analysis and results did not change.

4. in table 1, the percentage reported based on "row" as total, for example the proportion of male sex reported 92.2% in individuals W/O HTN, which is incorrect. this should be changed to "column" as total.

Response:

Thanks for mentioning. It is corrected in the manuscript. 

5. The conclusion should be according to the aim of the study, please add a sentence or two explaining the findings for best model.

Response:

Thank you for bringing up this point. We have included this statement in the conclusion section:

“LGBM emerged as the optimal model. It eventually introduced the top 30 features, highlighting the top 5 factors of higher baseline SBP, female gender, higher WHR, positive hematuria, and family history of hypertension significantly associated with hypertension development in the future. The model achieved an AUC of 0.67, f1-score=0.23 and AUC-PR=0.26.”

Reviewer #2

1. Abbreviations should be defined in their first use. Please ONLY use abbreviated forms after the definition (e.g., you defined ML several times).

Response:

Thanks for mentioning this point. The issue with the abbreviations has been fully corrected in the manuscript.

2. The introduction is too long. Make it more concise.

Response:

Thank you for your comment. We have revised the introduction section to make it a little shorter.

3. Line 116: change "5" to "five".

Response:

Thanks. It is corrected in the manuscript.

4. Methods section 2: Have you excluded patients receiving anti-hypertensive drugs?

Response:

We admire your accuracy. Yes, we excluded patients with hypertension who met the same diagnostic criteria mentioned in the final outcome section. It is now emphasized in the manuscript. Thanks.

“3. Participants without hypertension diseases at the first phase (with the same diagnostic criteria mentioned in the final outcome section)”

5. If possible, add external validation; else, mention it clearly in the discussion and limitations sections.

Response:

Due to limitations in accessing "full" datasets from different cohorts, we were unable to perform external validation. We acknowledge the significance of this comment and have included a statement addressing this limitation in our discussion section:

“Furthermore, we were unable to perform external validation with our models due to limitations in accessing complete datasets from different cohorts.”

6. I found several typos and grammatical errors.

Response:

Thank you for mentioning that. We have thoroughly reviewed the text and made the necessary corrections to fix any typos and grammatical errors.

Reviewer #3

1. Lines 85-87: Clarify the term "few risk factors" by providing a specific range. Additionally, elaborate on "common machine learning approaches" by offering examples from previous studies.

Response:

That's an important point. In contrast to many other studies, we included a large number of variables (up to 160) in our machine learning models. Although we did a comprehensive literature search and we are certain of this statement, providing the exact range of risk factors included in machine learning models from the existing literature is challenging, as we might inadvertently miss a study, resulting in an inaccurate range; Thus, we have fully omitted this sentence. 

“However, the data from these studies have been primarily cross-sectional, and there is no evidence indicating the successful implementation of these algorithms in clinical settings in the rural Middle East areas.”

2. Line 92: Introduce the acronym FACS before using it in this section for better reader understanding.

Response:

Thanks for the mentioned point. We have fully described FACS in the method section, so we have omitted it from the introduction section to avoid redundancy.

“In this investigation, we aim to assess and contrast the efficacy of various ML methods utilizing a longitudinal rural middle eastern dataset to forecast”

3. Lines 93-94: Distinguish between "common" and "established" machine learning techniques. Clarify how these terms differ and reconsider the use of "common" to avoid potential underestimation of previous approaches.

Response:

Thanks for mentioning this point. This is solely due to our limited English language proficiency. Based on your valuable comment, we have decided to remove the words "common" and "established" to avoid confusion and the underestimation of previous studies.

4. Line 108: Define the acronym NCDs before its use.

Response:

Thanks. We defined the abbreviation NCD

5. Line 108: Specify examples of "the most common ones" regarding NCDs for better context.

Response:

Thank you for bringing this up. We defined the abbreviation NCD and provided the most important example of NCDs present in our dataset and relate to our work. The inappropriate phrase of "the most common ones" is omitted.

“FACS was created to assess the risk factors that predispose Fasa's rural residents to Non-Communicable Diseases (NCDs), including cardiovascular diseases.”

6. Line 117: Reevaluate the necessity of explicitly stating "Age between 35-70 years" as an inclusion criterion, given that it aligns with the larger study's age range.

Response:

Thank you for pointing out this important mistake. Based on your valuable comment, we have removed the age range of 35-70 from our inclusion criteria.

7. Lines 123: Clarify the apparent conflict between the statement about missing data and the inclusion criterion.

Response:

Thanks for your accurate comment. We changed inclusion criteria number 3 (participants with complete data) to participants with 5 years data available.

“2. Participants with 5 years data available”

8. Lines 123-124: Provide details on how the multiple imputation method was implemented. Specify if different ML models were trained on various imputed datasets and how this influenced later stages.

Response:

Sorry for misunderstanding. We used 2 types of imputed methods and called this ‘multiple imputation’. We used mean and median imputation methods for continuous and categorical variables, respectively. We hadn’t multiple datasets in this approach and these was only one imputed data. We corrected the term ‘multiple imputation’ in the manuscript.

“For continuous variables, mean imputation was employed, while for categorical variables, median imputation was used to replace missing data.”

9. Lines 144-148: Explain the rationale behind choosing specific ML algorithms. Provide insights into why these algorithms were deemed suitable for the study.

Response:

We used a variety of ML methods to make sure the dataset was thoroughly explored. Every algorithm possesses distinct advantages and disadvantages, and our objective was to evaluate each one's performance independently in several research domains. SVM was selected for its capability in handling high-dimensional data and finding complex relationships. RF was leveraged as an ensemble learning model. GNB provided a computationally efficient approach. LDA offered interpretability. KNN was implemented to detect local patterns. GBM sequentially refined model performance. XGB can perform with high accuracy in large datasets. CAT optimized categorical feature handling, and LGBM efficiently managed larger datasets with swift training. This multifaceted strategy sought to capitalize on the distinct advantages of every model, guaranteeing a thorough examination of the dataset. By not depending only on a single model, we were able to prevent any bias and obtain a comprehensive comprehension of the data.

The whole paragraph mentioned above was added to the method (section 6).

10. Lines 156-158: Elaborate on your approach to combining hyperparameters. Explain whether a grid search or any specific method was used.

Response:

For the hyper-parameter tuning step, the grid search approach was employed. This sentence was added to the manuscript.

---

## [Decision Letter · Decision Letter 1]

23 Feb 2024

Machine learning-based models to predict the conversion of normal blood pressure to hypertension within 5-year follow-up

PONE-D-23-44016R1

Dear Dr. Tabrizi,

We’re pleased to inform you that your manuscript has been judged scientifically suitable for publication and will be formally accepted for publication once it meets all outstanding technical requirements.

Kind regards,

Amir Hossein Behnoush

Academic Editor

PLOS ONE

Additional Editor Comments (optional):

Reviewers' comments:

Reviewer's Responses to Questions

**Comments to the Author**

1. If the authors have adequately addressed your comments raised in a previous round of review and you feel that this manuscript is now acceptable for publication, you may indicate that here to bypass the “Comments to the Author” section, enter your conflict of interest statement in the “Confidential to Editor” section, and submit your "Accept" recommendation.

Reviewer #1: All comments have been addressed

Reviewer #2: All comments have been addressed

Reviewer #3: All comments have been addressed

2. Is the manuscript technically sound, and do the data support the conclusions?

Reviewer #1: Yes

Reviewer #2: (No Response)

Reviewer #3: Yes

3. Has the statistical analysis been performed appropriately and rigorously? 

Reviewer #1: Yes

Reviewer #2: (No Response)

Reviewer #3: Yes

4. Have the authors made all data underlying the findings in their manuscript fully available?

Reviewer #1: Yes

Reviewer #2: (No Response)

Reviewer #3: No

5. Is the manuscript presented in an intelligible fashion and written in standard English?

Reviewer #1: Yes

Reviewer #2: (No Response)

Reviewer #3: Yes

6. Review Comments to the Author

Reviewer #1: Many thanks for the precise responses. All comments are addressed properly and the manuscript meets acceptance criteria now.

Reviewer #2: (No Response)

Reviewer #3: (No Response)

7. PLOS authors have the option to publish the peer review history of their article (what does this mean?). If published, this will include your full peer review and any attached files.

Reviewer #1: No

Reviewer #2: No

Reviewer #3: No

---

## [Editor Report · Acceptance letter]

4 Mar 2024

PONE-D-23-44016R1 

PLOS ONE

Dear Dr. Tabrizi, 

I'm pleased to inform you that your manuscript has been deemed suitable for publication in PLOS ONE. Congratulations! Your manuscript is now being handed over to our production team.

Kind regards, 

on behalf of

Dr. Amir Hossein Behnoush 

Academic Editor

PLOS ONE